# Prevalence and correlates of long-term e-cigarette and nicotine replacement therapy use: a prospective study in England

Sarah E Jackson  ,[1] Emily Hill,[1] Lion Shahab,[1] Emma Beard,[1] Susan Michie,[2] Jamie Brown[1]

[1]Department of Behavioural Science and Health, University College London, London, United Kingdom
[2]Department of Clinical, Educational and Health Psychology, University College London, London, United Kingdom

**Correspondence to**
Dr Sarah E Jackson;
s.e.jackson@ucl.ac.uk

## ABSTRACT

**Objectives** To examine the prevalence of, and sociodemographic and smoking-related characteristics associated with, long-term e-cigarette use compared with long-term nicotine replacement therapy (NRT) use.

**Design** Cross-sectional and prospective survey, the Smoking Toolkit Study, with baseline data collected between September 2014 and September 2016 and follow-ups at 6 and 12 months.

**Setting** England.

**Participants** Population representative sample of 40 933 adults aged 16+ years.

**Main outcome measures** Prevalence of long-term (≥12 months) use of e-cigarettes and NRT by retrospective self-report among baseline respondents (all adults, n=40 933; smokers, n=8406) and current use at baseline, 6 months and 12 months in a subsample of smokers who responded to follow-up (n=733).

**Results** Of baseline respondents, 1.5% (95% CI 1.4% to 1.6%, n=604) of adults and 3.9% (95% CI 3.5% to 4.3%, n=327) of smokers were long-term e-cigarette users and 0.5% (95% CI 0.4% to 0.6%, n=205) of adults and 1.3% (95% CI 1.1% to 1.5%, n=112) of smokers were long-term NRT users. Assessed prospectively, 13.4% (95% CI 10.9% to 15.9%, n=100) of smokers were long-term e-cigarette users and 1.9% (95% CI 0.9% to 2.9%, n=14) were long-term NRT users. Among all adults, long-term use by never smokers of either e-cigarettes (0.1%, n=27) or NRT (0.0%, n=7) was rare. Among past-year smokers, long-term e-cigarette and NRT use was higher among older smokers compared with those who were 16–34 years old (OR range=1.55–5.21). Long-term e-cigarette use only was lower in smokers who were less educated (OR=0.63, 95% CI 0.49 to 0.81), from social grades C2DE (OR=0.66, 95% CI 0.52 to 0.84) and with children in the household (OR=0.66, 95% CI 0.51 to 0.85). Long-term e-cigarette use and long-term NRT use were higher among smokers more motivated to quit (OR=2.05, 95% CI 1.63 to 2.60 and OR=2.33, 95% CI 1.57 to 3.46).

**Conclusions** In the adult population in England, long-term use of e-cigarettes and long-term use of NRT are almost exclusively by current or ex-smokers. Only a minority of past-year smokers retrospectively report long-term e-cigarette or NRT use, but this figure may be an underestimate, especially for e-cigarette use, which is more than threefold higher when assessed prospectively.

---

### Strengths and limitations of this study

► Large sample representative of the adult population in England.
► Longitudinal design permitting prospective assessment of long-term use in addition to cross-sectional analyses based on retrospective self-reports.
► Only respondents who reported past-year smoking at baseline were invited to participate in the follow-up surveys, so we were unable to obtain prospective estimates of the prevalence of long-term e-cigarette or nicotine replacement therapy (NRT) use in the entire adult population.
► Substantial attrition bias meant our sample for prospective analyses was older and more socioeconomically advantaged than the group who were lost to follow-up and more reported recent quitting and long-term use of e-cigarettes or NRT retrospectively at baseline.

## INTRODUCTION

Tobacco smoking is one of the leading causes of premature death and disability worldwide.[1] The primary cause of smoking-associated morbidity and mortality is the inhalation of toxins produced from the combustion of tobacco.[2] Over recent years, electronic cigarettes (e-cigarettes) have rapidly become popular among smokers as a non-combustible alternative to cigarettes that offers safer nicotine delivery.[3] However, while the prevalence of ever and current use of e-cigarettes has been monitored (eg, refs 4–6), there has been little investigation into long-term use of these products. Given an increasing focus on harm reduction in tobacco control, which aims to reduce the harm from combustible products by partial or complete substitution with non-combustible products, high-quality data on long-term use are needed. Understanding who is using e-cigarettes, and for how long, is fundamental in order to evaluate their overall impact on public health.

In England, e-cigarettes are used by around 5% of the adult population (~20% of smokers)[3] and have overtaken nicotine replacement therapy (NRT) as the most popular quitting aid, with over a third of smokers using an e-cigarette in their most recent quit attempt compared with one in five using of NRT.[7] In England, e-cigarettes are not currently available on prescription but are subject to the EU Tobacco Products Directive (including advertising restrictions) and Trading Standards and can be bought online and from vape shops, pharmacies and other retail outlets, while NRT can be bought over the counter or obtained on prescription from a licenced health professional. Evidence from three randomised controlled trials indicates that using e-cigarettes in a quit attempt increases chances of successful cessation.[8 9] On a population level, the rise in use of e-cigarettes in England and the USA has been associated with increases in the overall success rate of quit attempts in the population,[10 11] likely contributing to continued declines in smoking prevalence.[12] It is possible that long-term e-cigarette use could help mitigate the high risk of relapse among recent quitters[13]; in a survey of US smokers with 2-year follow-up, long-term use of e-cigarettes (current use at baseline and follow-up) was associated with four times higher odds of cessation relative to no use.[14]

Accumulating evidence demonstrates that using e-cigarettes is substantially less harmful than smoking.[3] Toxicology testing has shown that while e-cigarettes can be used to obtain similar levels of nicotine to combustible cigarettes, switching to e-cigarettes can significantly reduce levels of measured carcinogens and toxins relative to smoking only combustible cigarettes, with differences observed within a matter of weeks.[15–17] A more favourable toxicity profile has also been observed among long-term e-cigarette users (≥6 months) compared with current cigarette smokers.[18] However, surveys have indicated that around half of smokers inaccurately judge e-cigarettes to be more harmful than combustible cigarettes, about as harmful, or are unsure about the relative risk,[19] which could discourage use.

Previous studies that have examined correlates of e-cigarette use have found that smokers who use e-cigarettes tend to be younger than non-users, smoke more heavily and are more likely to have tried to quit in the past year.[4–6] There is also some evidence that e-cigarette use is more prevalent among people with greater socioeconomic advantage,[5 6] although the gap appears to have narrowed over recent years.[20] However, there is a distinct lack of evidence on both the prevalence of long-term use and the profile of long-term users. This information is important for the evaluation of the overall public health impact of e-cigarettes, which requires specification of a wide variety of parameters beyond the safety of e-cigarettes and their effect on cessation, including the extent and characteristics of people who become long-term users.[21]

The present study therefore aimed to examine the prevalence of, and sociodemographic and smoking-related characteristics associated with, long-term (≥12 months) e-cigarette use in England. We also analysed data on long-term NRT use as a case–control, in order to assess the extent to which the prevalence of long-term e-cigarette use and profile of long-term users are specific to e-cigarettes or apply more broadly to non-combustible nicotine products in general. Specifically, we aimed to answer the following research questions:

1. What proportion of adults in England retrospectively report using (1) e-cigarettes or (2) NRT for at least 1 year?
2. What proportion of past-year smokers in England retrospectively report using (1) e-cigarettes or (2) NRT for at least 1 year?
3. What proportion of past-year smokers in England report current use of (1) e-cigarettes or (2) NRT at baseline and both 6-month and 12-month follow-ups?
4. How do long-term users of e-cigarettes and NRT differ from non-users in their sociodemographic and smoking-related characteristics?

## METHOD
### Design and study population
Data were used from the Smoking Toolkit Study (STS), an ongoing monthly repeat cross-sectional survey of adults in England.[22] Each month, a form of random location sampling is used to select a new sample of approximately 1700 adults aged 16 years and older. Grouped output areas (containing ~300 households) are stratified by ACORN (sociodemographic) characteristics (http://www.caci.co.uk/acron/acornmap.asp) and region before being randomly selected for inclusion in an interviewers list. Interviewers then choose which houses within these areas are most likely to fulfil their quotas and conduct face-to-face computer-assisted interviews with one member per household. Comparisons of sociodemographic data and smoking prevalence and consumption estimates with national data indicate that STS data are broadly representative of the English population, having a similar composition to other large national surveys, such as the Health Survey for England.[22] All participants provide fully informed consent prior to participation. In each wave, respondents complete a face-to-face computer-assisted survey with a trained interviewer. Respondents to the baseline survey between September 2014 and September 2016 who reported smoking in the past year were asked whether they were willing to be recontacted, and those who agreed were followed up by telephone 6 and 12 months after the baseline interview. Up to seven attempts were made to follow up each consenting participant. For the purpose of the present study, we aggregated data across survey waves. Cross-sectional analyses used data from all adults who responded to the baseline survey during this period, and the prospective analysis used data from respondents who reported past-year smoking at baseline and responded to both the 6-month and 12-month follow-up surveys.

## Patient and public involvement

The wider toolkit study has been discussed with a diverse patient and public involvement group, and the authors regularly attend and present at meetings at which patients and public are included. Interaction and discussion at these events help to shape the broad research priorities and questions. There is also a mechanism for generalised input from the wider public: each month interviewers seek feedback on the questions from all 1700 respondents, who are representative of the English population. This feedback is limited and usually simply relates to understanding of questions and item options. No patients or members of the public were involved in setting the research questions or the outcome measures, nor were they involved in the design and implementation of this specific study. There are no plans to involve patients in dissemination.

## Measures

### Outcomes: long-term use of e-cigarettes and NRT

The outcomes were long-term (≥12 months) use of e-cigarettes and long-term use of NRT, assessed retrospectively at baseline and prospectively over a 12-month follow-up.

In each of the baseline and follow-up surveys, three questions asked respondents about current use of e-cigarettes (or other vaping devices) and NRT (eg, nicotine patches, gum, spray or any other product):

1. Which, if any, of these are you currently using to help you cut down the amount you smoke?
2. Do you regularly use either of these in situations when you are not allowed to smoke?
3. Can I check, do you currently use either of the following at all for any reason?

In the baseline survey, respondents reporting use of either e-cigarettes or NRT were asked: 'How long have you been using this nicotine replacement product or these products for?' Response options were: (1) less than 1 week, (2) 1–6 weeks, (3) more than 6 weeks and up to 12 weeks, (4) more than 12 weeks and up to 26 weeks, (5) more than 26 weeks and up to 52 weeks and (6) more than 52 weeks.

For the present analyses, long-term use of e-cigarettes/NRT was defined as current use initiated more than 52 weeks prior to the baseline survey for cross-sectional analyses and as current use at baseline, 6 months and 12 months for prospective analyses. Participants who reported long-term use of both e-cigarettes and NRT (n=66) were excluded.

### Sociodemographic and smoking-related characteristics

Data were included on a range of sociodemographic and smoking-related characteristics assessed at baseline, selected a priori on the basis of previous studies demonstrating associations with use of e-cigarettes and/or NRT.

Sociodemographic information included: age, sex, ethnicity, region, social grade, level of education, disability and the presence of children in the household.

Ethnicity was categorised as white versus non-white. Region was defined according to Government Office Region, grouped into three categories: northern, central and southern England. Social grade was categorised as ABC1 (which includes managerial, professional and intermediate occupations) versus C2DE (which includes small employers and own-account workers, lower supervisory and technical occupations, and semiroutine and routine occupations, never workers and long-term unemployed). This occupational measure of social grade is a valid index of Socio-economic status (SES) that is widely used in research in UK populations. It has been identified as particularly relevant in the context of tobacco use and quitting[23] and other addictive behaviours.[24] These social grades are frequently amalgamated into two groupings: ABC1 and C2DE. Here, researchers frequently interpret ABC1 to represent the middle class and C2DE to represent the working class. Education was categorised as lower (no post-16 qualifications) versus higher (higher level qualifications above GCSE level). Disability status was identified from the question 'Do you consider yourself to have a disability within the meaning of the Disability Discrimination Act 1995 (yes/no)?'. The number of children in the household was self-reported and dichotomised to 0 versus ≥1.

Smoking-related characteristics included: smoking status, time to first cigarette, consistent motivation to stop and (because it has been shown to be associated with smoking and quitting behaviour[25–27]) high-risk drinking. Smoking status was self-reported by all adults in response to the question: 'Which of the following best applies to you (current smoker/stopped in the past year/stopped more than a year ago/never smoked)?'. Respondents who reported current smoking or having stopped in the past year ('past-year smokers') were also asked how soon after waking they typically smoked their first cigarette (categorised as within 30 vs ≥31 min; an established indicator of nicotine dependence[28]) and whether they had consistently felt that they wanted to stop in the past year (yes/no). High-risk drinking was assessed using the Alcohol Use Disorders Identification Test,[29] a 10-item screening tool developed by the WHO to assess alcohol consumption, drinking behaviours and alcohol dependence, with a score of 8 or more indicating high-risk drinking.

### Statistical analysis

The analysis plan and syntax were preregistered on Open Science Framework (https://osf.io/bpjhk/). All analyses were done in SPSS V.25 on complete cases.

We used $\chi^2$ tests to compare the baseline characteristics of individuals who responded to both the 6-month and 12-month follow-ups with those who were lost to follow-up in order to assess the representativeness of those followed up.

We estimated the weighted prevalence of long-term use of (1) e-cigarettes and (2) NRT in the total adult population at baseline and in past-year smokers at baseline and over 12-month follow-up. Rim (marginal) weighting was

used to match the English population on the dimensions of age, social grade, region, housing tenure (bought on a mortgage, owned outright, rented from local authority and rented from private landlord), ethnicity and working status (working or not working) within sex.

We then used logistic regression to examine the extent to which sociodemographic and smoking-related characteristics were associated with long-term use of e-cigarettes and NRT, assessed at baseline. For each outcome, we analysed bivariate associations with each potential correlate separately and tested independent associations with a multivariable model that included all variables. We had also intended to analyse associations with long-term use prospectively, but the achieved sample size was lower than we had anticipated, and the prevalence of long-term use was low (particularly for NRT), limiting statistical power.

Following peer review, we added an unplanned sensitivity analysis of the prospective data in which missing data on e-cigarette and NRT use at 6 months and 12 months were imputed for all baseline past-year smokers with missing data. We used a multiple imputation model with all baseline sociodemographic and smoking-related characteristics, baseline use of e-cigarettes and baseline use of NRT as predictors. Five imputed datasets were created, each analysed separately, and the results combined to produce pooled estimates of prevalence of long-term use of e-cigarettes and long-term use of NRT.

## RESULTS
### Long-term use of e-cigarettes and NRT among all adults in England: cross-sectional data
A total of 42 040 adults in England were surveyed between September 2014 and September 2016, and 40 933 (97.4%) were complete cases. The weighted prevalence of long-term e-cigarette use assessed retrospectively among all adults in England was 1.5% (95% CI 1.4% to 1.6%) and of long-term NRT use was 0.5% (95% CI 0.4% to 0.6%). table 1 summarises sample characteristics and bivariate and multivariable associations between sociodemographic and smoking-related characteristics and long-term use of e-cigarettes and NRT among all adults in the baseline survey.

In the multivariable model, both long-term e-cigarette use and long-term NRT use were significantly associated with age, region, level of education, disability and smoking status. Compared with those aged 16–34 years, long-term e-cigarette use was more prevalent among those aged 35–54 years but was not significantly different among those aged ≥55 years. Long-term NRT was significantly more prevalent among those aged 35–54 and ≥55 years. Compared with the north of England, long-term e-cigarette use was less prevalent in central and southern regions, and long-term NRT use was more prevalent in the south. Both long-term e-cigarette use and long-term NRT use were significantly less prevalent among people with no post-16 qualifications and more prevalent among those with a disability. Prevalence of long-term e-cigarette

and NRT use did not differ significantly between current smokers and recent ex-smokers but was significantly less prevalent among never smokers, among whom use of either product was rarely reported (e-cigarettes 0.1%, NRT 0.0%). Long-term NRT use was also significantly less prevalent among long-term ex-smokers, but the prevalence of long-term e-cigarette use did not differ significantly between long-term ex-smokers and current smokers.

Long-term NRT use, but not long-term e-cigarette use, was associated with sex and high-risk drinking, with higher prevalence observed among women and high-risk drinkers. Long-term e-cigarette use, but not long-term NRT use, was associated with the presence of children in the household, with lower prevalence observed among people with children in their household. We observed no significant association between long-term e-cigarette or NRT use and ethnicity or social grade.

### Long-term use of e-cigarettes and NRT among past-year smokers in England: cross-sectional data
A total of 8649 participants were past-year smokers and 8406 (97.2%) were complete cases. The weighted prevalence of long-term e-cigarette use assessed retrospectively among past-year smokers in England was 3.9% (95% CI 3.5% to 4.3%) and of long-term NRT use was 1.3% (95% CI 1.1% to 1.5%). table 2 summarises bivariate and multivariable associations between sociodemographic and smoking-related characteristics and long-term use of e-cigarettes and NRT among past-year smokers in the baseline survey.

In the multivariable model, there were significant associations between long-term use of e-cigarettes and NRT and age, region and motivation to stop smoking, and between long-term use of e-cigarettes and social grade, level of education and children in the household. Long-term use of e-cigarettes and NRT was more prevalent among older smokers compared with 16–34 years old, and among those who were motivated to stop. Compared with the north of England, long-term e-cigarette use was less prevalent in central and southern regions, but long-term NRT use was more prevalent in the south. Long-term e-cigarette use was significantly less prevalent among smokers from social grades C2DE, without post-16 qualifications and with children in their household, while long-term NRT use did not differ significantly according to these factors. We observed no significant association between long-term e-cigarette or NRT use and sex, ethnicity, disability, current smoking status, excessive drinking or dependence.

### Long-term use of e-cigarettes and NRT among past-year smokers in England: prospective data
A total of 733 individuals who reported past-year smoking at baseline completed follow-up surveys at both 6 and 12 months. Characteristics of past-year smokers in the baseline and follow-up samples are summarised in table 3. Past-year smokers who responded to follow-up were

**Table 1** Sample descriptive characteristics and factors associated with long-term use of e-cigarettes or NRT by adults in England: cross-sectional data (n=40 933)

| | Whole sample | E-cigarettes | | | NRT | | |
|---|---|---|---|---|---|---|---|
| | % (n) | % (n) long-term use | OR (95% CI) p value | Adj OR (95% CI) p value | % (n) long-term use | OR (95% CI) p value | Adj OR (95% CI) p value |
| All adults* | – | 1.5 (604) | – | – | 0.5 (205) | – | – |
| **Age in years** | | | | | | | |
| 16–34 | 30.3 (12 398) | 1.1 (138) | 1.00 | 1.00 | 0.2 (21) | 1.00 | 1.00 |
| 35–54 | 29.6 (12 118) | 2.1 (252) | 1.89 (1.53 to 2.33) <0.001 | 1.71 (1.38 to 2.13) <0.001 | 0.7 (87) | 4.26 (2.65 to 6.87) <0.001 | 3.98 (2.45 to 6.47) <0.001 |
| ≥55 | 40.1 (16 417) | 1.1 (185) | 1.01 (0.81 to 1.26) 0.912 | 0.78 (0.60 to 1.02) 0.064 | 0.6 (89) | 3.21 (2.00 to 5.17) <0.001 | 3.25 (1.91 to 5.52) <0.001 |
| **Sex** | | | | | | | |
| Men | 50.9 (20 816) | 1.5 (308) | 1.00 | 1.00 | 0.5 (94) | 1.00 | 1.00 |
| Women | 49.1 (20 117) | 1.3 (267) | 0.90 (0.76 to 1.06) 0.190 | 1.04 (0.87 to 1.23) 0.683 | 0.5 (103) | 1.14 (0.86 to 1.50) 0.377 | 1.39 (1.04 to 1.86) 0.025 |
| **Ethnicity** | | | | | | | |
| Non-white | 16.4 (6730) | 0.6 (42) | 1.00 | 1.00 | 0.1 (10) | 1.00 | 1.00 |
| White | 83.6 (34 203) | 1.6 (533) | 2.52 (1.84 to 3.46) <0.001 | 1.12 (0.80 to 1.56) 0.510 | 0.5 (187) | 3.69 (1.95 to 6.98) <0.001 | 1.48 (0.77 to 2.86) 0.240 |
| **Social grade** | | | | | | | |
| ABC1 | 53.5 (21 894) | 1.3 (292) | 1.00 | 1.00 | 0.5 (102) | 1.00 | 1.00 |
| C2DE | 46.5 (19 039) | 1.5 (283) | 1.12 (0.95 to 1.32) 0.191 | 0.91 (0.76 to 1.10) 0.326 | 0.5 (95) | 1.07 (0.81 to 1.42) 0.629 | 0.93 (0.68 to 1.27) 0.644 |
| **Region** | | | | | | | |
| North | 32.0 (13 111) | 2.1 (281) | 1.00 | 1.00 | 0.4 (57) | 1.00 | 1.00 |
| Central | 29.3 (12 000) | 1.1 (132) | 0.51 (0.41to 0.63)<0.001 | 0.60 (0.48 to 0.74) <0.001 | 0.5 (59) | 1.13 (0.79 to 1.63) 0.507 | 1.41 (0.97 to 2.04) 0.069 |
| South | 38.7 (15 822) | 1.0 (162) | 0.47 (0.39 to 0.57)<0.001 | 0.55 (0.45 to 0.67) <0.001 | 0.5 (81) | 1.18 (0.84 to 1.66) 0.343 | 1.45 (1.02 to 2.04) 0.036 |
| **Level of education** | | | | | | | |
| Post-16 qualifications | 63.4 (25 945) | 1.4 (371) | 1.00 | 1.00 | 0.5 (126) | 1.00 | 1.00 |
| No post-16 qualifications | 36.6 (14 988) | 1.4 (204) | 0.95 (0.80 to 1.13) 0.569 | 0.79 (0.65 to 0.95) 0.012 | 0.5 (71) | 0.98 (0.73 to 1.31) 0.867 | 0.71 (0.52 to 0.98) 0.036 |
| **Disability** | | | | | | | |
| No | 88.2 (36 119) | 1.3 (463) | 1.00 | 1.00 | 0.4 (152) | 1.00 | 1.00 |

Continued

**Table 1** Continued

| | Whole sample | E-cigarettes | | | NRT | | |
|---|---|---|---|---|---|---|---|
| Yes | 11.8 (4814) | 2.3 (112) | 1.83 (1.49 to 2.26) <0.001 | 1.34 (1.07 to 1.66) 0.009 | 0.9 (45) | 2.23 (1.60 to 3.12) <0.001 | 1.52 (1.08 to 2.15) 0.017 |
| Children in the household | | | | | | | |
| 0 | 71.0 (29 062) | 1.4 (413) | 1.00 | 1.00 | 0.5 (145) | 1.00 | 1.00 |
| ≥1 | 29.0 (11 871) | 1.4 (162) | 0.96 (0.80 to 1.15) 0.660 | 0.80 (0.65 to 0.98) 0.030 | 0.4 (52) | 0.88 (0.64 to 1.21) 0.420 | 1.00 (0.69 to 1.45) 0.998 |
| Current smoking status | | | | | | | |
| Current smoker | 19.3 (7900) | 3.8 (302) | 1.00 | 1.00 | 1.4 (107) | 1.00 | 1.00 |
| Recent (<1 year) ex-smoker | 1.4 (558) | 3.8 (21) | 0.98 (0.63 to 1.54) 0.944 | 0.93 (0.59 to 1.46) 0.747 | 1.3 (7) | 0.93 (0.43 to 2.00) 0.843 | 0.87 (0.40 to 1.89) 0.724 |
| Long-term (≥1 year) ex-smoker | 17.1 (7009) | 3.2 (225) | 0.83 (0.70 to 1.00) 0.043 | 0.90 (0.74 to 1.09) 0.274 | 1.1 (76) | 0.80 (0.59 to 1.07) 0.136 | 0.62 (0.45 to 0.85) 0.003 |
| Never smoker | 62.2 (25 466) | 0.1 (27) | 0.03 (0.02 to 0.04) <0.001 | 0.03 (0.02 to 0.04) <0.001 | 0.0 (7) | 0.02 (0.01 to 0.04) <0.001 | 0.02 (0.01 to 0.04) <0.001 |
| High-risk drinking | | | | | | | |
| No | 87.3 (35 742) | 1.3 (449) | 1.00 | 1.00 | 0.4 (149) | 1.00 | 1.00 |
| Yes | 12.7 (5191) | 2.4 (126) | 1.95 (1.60 to 2.39) <0.001 | 1.02 (0.83 to 1.27) 0.830 | 0.9 (48) | 2.22 (1.61 to 3.09) <0.001 | 1.60 (1.13 to 2.26) 0.008 |

The adjusted model includes all variables in the table and year of survey.

*This figure is weighted, and therefore the effective N does not correspond precisely with the unweighted figures reported elsewhere in the table.

Adj OR, adjusted OR; NRT, nicotine replacement therapy.

**Table 2** Factors associated with long-term use of e-cigarettes or NRT by past-year smokers in England: cross-sectional data (n=8406)

| | E-cigarettes | | | NRT | | |
|---|---|---|---|---|---|---|
| | % (n) long-term use | OR (95% CI) p value | Adj OR (95% CI) p value | % (n) long-term use | OR (95% CI) p value | Adj OR (95% CI) p value |
| All adults* | 3.9 (327) | – | – | 1.3 (112) | – | – |
| **Age in years** | | | | | | |
| 16–34 | 2.6 (88) | 1.00 | 1.00 | 0.5 (15) | 1.00 | 1.00 |
| 35–54 | 4.9 (135) | 1.88 (1.43 to 2.47) <0.001 | 1.90 (1.44 to 2.52) <0.001 | 1.7 (46) | 3.72 (2.07 to 6.67) <0.001 | 3.47 (1.92 to 6.27) <0.001 |
| ≥55 | 4.3 (98) | 1.64 (1.22 to 2.19) 0.001 | 1.55 (1.12 to 2.13) 0.008 | 2.3 (52) | 5.10 (2.86 to 9.08) <0.001 | 5.21 (2.79 to 9.72) <0.001 |
| **Sex** | | | | | | |
| Men | 4.0 (170) | 1.00 | 1.00 | 1.3 (57) | 1.00 | 1.00 |
| Women | 3.8 (157) | 0.96 (0.77 to 1.20) 0.702 | 1.00 (0.80 to 1.27) 0.974 | 1.4 (56) | 1.15 (0.79 to 1.66) 0.474 | 1.14 (0.78 to 1.68) 0.498 |
| **Ethnicity** | | | | | | |
| Non-white | 3.4 (31) | 1.00 | 1.00 | 0.9 (8) | 1.00 | 1.00 |
| White | 3.9 (290) | 1.15 (0.79 to 1.68) 0.460 | 1.01 (0.68 to 1.50) 0.961 | 1.4 (105) | 1.62 (0.79 to 3.33) 0.192 | 1.33 (0.63 to 2.80) 0.459 |
| **Social grade** | | | | | | |
| ABC1 | 5.2 (167) | 1.00 | 1.00 | 1.7 (54) | 1.00 | 1.00 |
| C2DE | 3.0 (154) | 0.56 (0.45 to 0.70) <0.001 | 0.66 (0.52 to 0.84) 0.001 | 1.1 (58) | 0.65 (0.45 to 0.94) 0.022 | 0.71 (0.47 to 1.06) 0.091 |
| **Region** | | | | | | |
| North | 4.7 (147) | 1.00 | 1.00 | 1.0 (32) | 1.00 | 1.00 |
| Central | 3.4 (81) | 0.73 (0.55 to 0.96) 0.023 | 0.79 (0.59 to 1.04) 0.094 | 1.4 (33) | 1.38 (0.85 to 2.25) 0.197 | 1.52 (0.92 to 2.50) 0.100 |
| South | 3.2 (93) | 0.68 (0.52 to 0.88) 0.004 | 0.68 (0.52 to 0.90) 0.006 | 1.7 (48) | 1.64 (1.05 to 2.58) 0.030 | 1.76 (1.11 to 2.79) 0.016 |
| **Level of education** | | | | | | |
| Post-16 qualifications | 4.7 (214) | 1.00 | 1.00 | 1.4 (65) | 1.00 | 1.00 |
| No post-16 qualifications | 2.7 (107) | 0.57 (0.45 to 0.72) <0.001 | 0.63 (0.49 to 0.81) <0.001 | 1.2 (48) | 0.85 (0.59 to 1.24) 0.411 | 0.85 (0.56 to 1.27) 0.426 |
| **Disability** | | | | | | |
| No | 3.7 (260) | 1.00 | 1.00 | 1.2 (86) | 1.00 | 1.00 |
| Yes | 4.7 (61) | 1.29 (0.97 to 1.72) 0.079 | 1.22 (0.90 to 1.64) 0.197 | 2.1 (27) | 1.73 (1.12 to 2.67) 0.014 | 1.46 (0.93 to 2.29) 0.103 |
| **Children in the household** | | | | | | |
| 0 | 4.3 (240) | 1.00 | 1.00 | 1.5 (83) | 1.00 | 1.00 |
| ≥1 | 2.9 (81) | 0.66 (0.51 to 0.85) 0.001 | 0.68 (0.51 to 0.90) 0.007 | 1.1 (30) | 0.71 (0.47 to 1.08) 0.111 | 1.07 (0.66 to 1.72) 0.795 |

Continued

**Table 2** Continued

| | E-cigarettes | | | NRT | | |
|---|---|---|---|---|---|---|
| **Current smoking status** | | | | | | |
| Current smoker | 3.8 (301) | 1.00 | 1.00 | 1.3 (106) | 1.00 | 1.00 |
| Recent (<1 year) ex-smoker | 3.7 (20) | 0.96 (0.61 to 1.52) 0.858 | 0.69 (0.43 to 1.11) 0.124 | 1.3 (7) | 0.95 (0.44 to 2.06) 0.904 | 0.69 (0.32 to 1.52) 0.361 |
| **High-risk drinking** | | | | | | |
| No | 3.7 (242) | 1.00 | 1.00 | 1.3 (84) | 1.00 | 1.00 |
| Yes | 4.2 (79) | 1.14 (0.88 to 1.48) 0.323 | 1.02 (0.78 to 1.35) 0.871 | 1.5 (29) | 1.20 (0.79 to 1.84) 0.396 | 1.52 (0.97 to 2.38) 0.065 |
| **Time to first cigarette** | | | | | | |
| 31 or more minutes | 3.8 (163) | 1.00 | 1.00 | 1.2 (53) | 1.00 | 1.00 |
| Within 30 min | 3.9 (158) | 1.02 (0.82 to 1.28) 0.833 | 1.07 (0.85 to 1.35) 0.581 | 1.5 (60) | 1.20 (0.83 to 1.74) 0.341 | 1.16 (0.79 to 1.70) 0.447 |
| **Consistent motivation to stop** | | | | | | |
| No | 2.6 (123) | 1.00 | 1.00 | 0.9 (41) | 1.00 | 1.00 |
| Yes | 5.4 (198) | 2.11 (1.68 to 2.66) <0.001 | 2.05 (1.63 to 2.60) <0.001 | 1.9 (72) | 2.27 (1.54 to 3.33) <0.001 | 2.33 (1.57 to 3.46) <0.001 |

The adjusted model includes all variables in the table and year of survey.
*This figure is weighted, and therefore, the effective N does not correspond precisely with the unweighted figures reported elsewhere in the table.
Adj OR, adjusted OR; NRT, nicotine replacement therapy.

**Table 3** Comparison of the baseline and follow-up samples of past-year smokers

| | Baseline sample*<br>% (n=8406) | Follow-up sample<br>% (n=733) | P value† |
|---|---|---|---|
| Age in years | | | |
| 16–34 | 39.6 (3326) | 15.6 (114) | <0.001 |
| 35–54 | 33.0 (2777) | 34.4 (252) | – |
| ≥55 | 27.4 (2303) | 50.1 (367) | – |
| Women | 46.2 (3885) | 44.9 (329) | 0.449 |
| White ethnicity | 89.1 (7488) | 95.4 (699) | <0.001 |
| Social grade C2DE | 61.8 (5193) | 51.3 (376) | <0.001 |
| Region | | | |
| North | 37.5 (3150) | 39.4 (289) | 0.137 |
| Central | 28.1 (2363) | 25.0 (183) | – |
| South | 34.4 (2893) | 35.6 (261) | – |
| No post-16 qualifications | 46.3 (3893) | 40.4 (296) | 0.001 |
| Has a disability | 15.5 (1304) | 22.6 (166) | <0.001 |
| ≥1 children in the household | 33.6 (2826) | 22.9 (168) | <0.001 |
| Current smoking status | | | |
| Current smoker | 93.5 (7862) | 91.7 (672) | 0.033 |
| Recent (<1 year) ex-smoker | 6.5 (544) | 8.3 (61) | – |
| High-risk drinking | 22.4 (1879) | 20.1 (147) | 0.118 |
| First cigarette within 30 min | 48.6 (4089) | 50.3 (369) | 0.336 |
| Consistent motivation to stop | 43.9 (3694) | 48.6 (356) | 0.008 |
| Long-term e-cigarette use† | 3.8 (321) | 5.5 (40) | 0.015 |
| Long-term NRT use† | 1.3 (113) | 2.6 (19) | 0.002 |

*Past-year smokers only.
†Comparison of respondents who did and did not provide follow-up data.
‡Assessed at baseline.

significantly older than those who did not. A higher proportion of responders were white and fewer were from social grades C2DE or had no post-16 qualifications. More reported a disability and fewer had children in the household. A higher proportion of responders than non-responders were recent ex-smokers and more reported consistent motivation to stop smoking. They were also significantly more likely to report long-term use of e-cigarettes or NRT than those who did not respond to the follow-up surveys. Loss to follow-up was not significantly associated with sex, region, high-risk drinking or dependence. The weighted prevalence of long-term e-cigarette use assessed prospectively among past-year smokers in England was 13.4% (95% CI 10.9% to 15.9%) and of long-term NRT use was 1.9% (95% CI 0.9% to 2.9%).

When missing data on use of e-cigarettes and NRT at 6 and 12 months were multiply imputed for participants who were past-year smokers at baseline and did not participate in the follow-up surveys (n=1673, 69.5% of all baseline past-year smokers), the unweighted prevalence of long-term e-cigarette use assessed prospectively was 9.8% (95% CI 9.2% to 10.4%) and of long-term NRT use was 1.7% (95% CI 1.4% to 2.0%), and the weighted

prevalence was 10.3% (95% CI 9.7% to 11.0%) and 1.6% (95% CI 1.3% to 1.9%), respectively.

## DISCUSSION
In this large, representative sample of adults in England, long-term use of e-cigarettes and NRT was almost exclusively reported by current or ex-smokers. Only a minority of past-year smokers retrospectively reported long-term use of either e-cigarettes (3.9%) or NRT (1.3%), but this figure may be an underestimate: prevalence of current use at three time-points over a 12-month period was substantially higher for both e-cigarettes (13.4%) and NRT (1.9%), although these estimates were likely subject to attrition bias. When missing data were imputed, prospectively assessed prevalence estimates were slightly lower, at 10.3% for long-term e-cigarette use and 1.6% for long-term NRT use. Both cross-sectionally and prospectively, there was a higher prevalence of long-term e-cigarette use in comparison with NRT use. In adjusted models, long-term use of e-cigarettes and NRT was higher among older smokers and those more motivated to quit smoking. Long-term use of e-cigarettes was less common, and long-term

use of NRT was more common, in the south of England compared with the north. Long-term use of e-cigarettes was significantly less prevalent among smokers who were less educated, those from social grades C2DE, and those with children in the household, but these variables were not significantly associated with long-term use of NRT. Neither long-term use of e-cigarettes nor NRT differed significantly according to sex, ethnicity, disability, current smoking status (current vs recent ex-smokers), high-risk drinking or nicotine dependence.

To our knowledge, this study is the first to examine the prevalence of, and sociodemographic and smoking-related characteristics associated with, long-term use of e-cigarettes. We aimed to identify the prevalence of long-term e-cigarette use cross-sectionally and prospectively and to contrast usage with long-term NRT use. Our results showed a higher prevalence of long-term e-cigarette use in comparison with NRT. Recent prevalence estimates indicate that current use of e-cigarettes is much more popular than NRT[7] and the same appears true for long-term use.

Long-term use of both products was almost exclusively observed among current and former smokers. Concerns have been raised that e-cigarettes may serve as a gateway to cigarette smoking among never-smokers, particularly among youth,[30 31] but in our sample, the prevalence of long-term e-cigarette use among never-smokers was just 0.1%, comparable with long-term NRT use. As such, the potential number of people susceptible to any gateway effects as a result of long-term e-cigarette use in England between 2014 and 2016 appears to have been very small. We also observed higher prevalence of long-term e-cigarette use among middle-aged and older adults than in the youngest group (16–34 years), in contrast with evidence that current use of e-cigarettes among current and former smokers in England is least prevalent in the oldest age group (12.2% in those aged ≥65 years, compared with 18.7%, 21.4%, 20.8%, 20.6% and 18.6% in those aged 16–24, 25–34, 35–44, 45–54 and 55–64 years).[7]

While the prevalence of long-term use of e-cigarettes and NRT did not differ significantly between current and recent (<1 year) ex-smokers, the relative prevalence of use in long-term (≥1 year) ex-smokers differed between the products. Long-term use of NRT was significantly less prevalent among long-term ex-smokers than current smokers, whereas long-term use of e-cigarettes was similar between these groups. This suggests that people tend to discontinue use of NRT more quickly after quitting smoking than with e-cigarettes, possibly because e-cigarettes are a closer substitute for the behaviour of cigarette smoking than NRT or because NRT is viewed more as a medication than a recreational product.[9] A recent trial of e-cigarettes compared with NRT in UK stop smoking services observed similar, with participants randomised to use e-cigarettes in a quit attempt more likely than those randomised to use NRT to still be using their allocated product 1 year later (80% vs 9%, respectively).[9]

Insofar that use of alternative nicotine products should promote cessation rather than continued dual use, it is somewhat concerning that long-term use of e-cigarettes was similarly prevalent among current and recent ex-smokers. However, this appeared equally true for NRT. There have been concerns that dual use of cigarettes and e-cigarettes could reduce the urgency to quit smoking[32] and extend the duration of cigarette smoking.[33 34] This would result in a negative overall public health impact, since duration of smoking poses a greater health risk than intensity of smoking.[35] However, our results indicate that this is not the case: after mutual adjustment, the recall of long-term use of both e-cigarettes and NRT was higher among smokers who were more motivated to quit. This finding is consistent with previous studies showing that the most common reason for using an e-cigarette is to stop smoking,[3] that smokers who use e-cigarettes are more likely to have recently tried to quit[4–6] and that long-term e-cigarette use is associated with a higher rate of smoking cessation.[14] It suggests that long-term dual use is not associated with reduced motivation to quit.

Among past-year smokers, long-term use of e-cigarettes specifically was lower among those without post-16 qualifications and those from social grades C2DE. This is consistent with a larger literature on the diffusion of innovation, which recognises the tendency for high status groups to most quickly adopt new ideas and behaviours[36–38]; a pattern that was documented for combustible cigarette smoking.[39] According to this theory, one would expect to see e-cigarette use spread first within more affluent social networks, but patterns of imitation later lead to diffusion of the practice and normative change across the socioeconomic spectrum. The fact that we observed no significant association between education or social grade and long-term NRT use, which has a similar cost to users[40] but has been available for much longer, is consistent with this. With recent evidence indicating that the socioeconomic gradient in e-cigarette use is declining over time,[20] we predict that this disparity in long-term use will disappear over the coming years. There were also regional differences, with long-term e-cigarette use more prevalent in the north of England and long-term NRT use more prevalent in the south. The higher prevalence of long-term e-cigarette use in the north is consistent with previous evidence of heavier smoking in the north of England[41] and higher prevalence of e-cigarette use among heavier smokers.[4–6]

Prospective analysis of current use at baseline and both 6-month and 12-month follow-ups indicated a substantially higher rate of long-term use of both e-cigarettes (13.4%) and NRT (1.9%) among past-year smokers than was seen in the cross-sectional results. While these figures are not directly comparable given the substantial attrition over follow-up assessments and differences in the definition of long-term use (continued use vs current use at three defined time points), the magnitude of the difference indicates that retrospective recall of how long the products have been used may underestimate what could be expected if users were followed more frequently over

time. A study with a greater number of follow-up points over a longer period could offer further insight into this discrepancy.

Strengths of this study include the large, representative sample and prospective design. However, there were several limitations. Only respondents who reported past-year smoking at baseline were invited to participate in the follow-up surveys, so we were unable to obtain prospective estimates of the prevalence of long-term e-cigarette or NRT use in the entire adult population. While evidence from the cross-sectional results of this study and from previous research[3] suggest that the vast majority of long-term users were current or recent ex-smokers, with low prevalence among never smokers, it would have been useful to have data from long-term ex-smokers. Another potential issue was substantial attrition bias. Our sample for prospective analyses was older and more socioeconomically advantaged than the group who were lost to follow-up, and more reported long-term use of e-cigarettes or NRT retrospectively at baseline. They were also more likely to have quit recently. This may have contributed to the higher prevalence of long-term use observed in prospective analyses. Finally, we did not consider reasons for or patterns of use. Future studies could build on our findings through more detailed or frequent assessments and qualitative work with long-term users.

## Conclusions

Long-term e-cigarette use is more prevalent than long-term NRT use in the English adult population, but both are almost exclusively by current or ex-smokers. The profile of long-term e-cigarette users is broadly similar to that of long-term NRT users, although there are some sociodemographic and regional differences between the two, with long-term e-cigarette use more prevalent among smokers with greater socioeconomic advantage and in the north of England and long-term NRT use more prevalent among smokers in the south. Prospective assessment of long-term use produces substantially higher estimates of prevalence, particularly for e-cigarettes, than retrospective recall, although this may to some extent be accounted for by differences in the sample and definitions used. These results add to the descriptive picture of e-cigarette use in England, providing novel insight into long-term use. This information can be incorporated into broader evaluations of population-level use of e-cigarettes and their potential impact on public health.

**Author contributions** JB and EH conceived of the study. SEJ, EH and JB analysed and interpreted the data. SEJ and EH drafted the manuscript. All authors critically revised the manuscript for important intellectual content.

**Funding** This work was supported by Cancer Research UK, grant numbers C1417/A22962 and C44576/A19501. All researchers listed as authors are independent from the funders, and all final decisions about the research were taken by the investigators and were unrestricted.

**Disclaimer** The funders had no final role in the study design; in the collection, analysis and interpretation of data; in the writing of the report; or in the decision to submit the paper for publication.

**Competing interests** JB received unrestricted research funding from Pfizer, which manufacture smoking cessation medications. LS has received honoraria for talks, an unrestricted research grant and travel expenses to attend meetings and workshops from Pfizer, and has acted as paid reviewer for grant awarding bodies and as a paid consultant for healthcare companies. All authors declare no financial links with tobacco companies or e-cigarette manufacturers or their representatives.

**Patient consent for publication** Not required.

**Ethics approval and consent to participate** Ethical approval for the Smoking Toolkit Study was granted originally by the University College London (UCL) Ethics Committee (ID 0498/001), and participants provided full informed consent. The data are not collected by UCL and are anonymised when received by UCL.

**Provenance and peer review** Not commissioned; externally peer reviewed.

**Data availability statement** The datasets used and/or analysed during the current study are available from the corresponding author on reasonable request.

**ORCID iD**
Sarah E Jackson http://orcid.org/0000-0001-5658-6168

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
