## [Reviewer comments · BMJ Open]

ARTICLE DETAILS

TITLE (PROVISIONAL)	Prevalence and correlates of long-term e-cigarette and nicotine replacement therapy use: a prospective study in England
AUTHORS	Jackson, Sarah; Hill, Emily; Shahab, Lion; Beard, Emma; Michie, Susan; Brown, Jamie

VERSION 1 – REVIEW

REVIEWER	Ray Niaura New York University, USA
REVIEW RETURNED	24-Feb-2019

GENERAL COMMENTS	The authors are to be commended for a well-written study that anticipates important questions regarding long-term use of e-cigarette-type products (ecigs). They compare use of these products to long-term use of NRT, which provides information about enduring use of nicotine that is not delivered via tobacco. The study is descriptive in nature and the results tell us that long-term ecig use is not (yet) highly prevalent and that, importantly, their use is confined to current and ex-smokers. This is reassuring, and indicates that long-term use may confer some advantages in terms of helping smokers to cut down/quit combustible tobacco use. Minor point: The authors should describe in a bit more detail why the STS yields a representative sample that permits population-level statistical inferences.
--

REVIEWER	Sara Kalkhoran Massachusetts General Hospital Boston, Massachusetts, USA
REVIEW RETURNED	25-Feb-2019

GENERAL COMMENTS	This manuscript evaluates the prevalence of long-term e-cigarette and nicotine replacement therapy use among adults in England, and factors associated with long-term use of each product. The authors found a higher prevalence of long-term e-cigarette use than NRT use, both among all adults and among current smokers. They defined long-term product use in two different ways – retrospectively among all participants and prospectively among smokers – and found higher prevalence rates with the prospective definition, although the latter estimation had the limitation of potential bias from loss to follow-up and a difference in definition. Strengths of this study include the large sample size and representative sample. Limitations involve contextual factors as noted below. I do think that the paper would be strengthened by a comparison between long-term users of e-cigarettes/NRT and short-term users of e-cigarettes/NRT in this specific population for
--

	context; based on how e-cigarette use was assessed in the survey, it seems that this should be possible. Abstract, Setting: Would include the year(s) of data collection here. Abstract, Results: The reference to “Long-term e-cigarette and NRT use” sounds like it is referring to people who use both e-cigarettes and NRT. Would recommend rephrasing. Introduction, General: It would be helpful to have some context on the regulatory/marketing environment in the England for non-UK readers. Are both e-cigarettes and NRT available over the counter? Are they available by prescription? I.e. how easy is it to access each product Introduction, Page 4, Paragraph 2: Would add the 2019 RCT in NEJM by Hajek et al to the two RCTs mentioned here. Introduction, Page 4, Paragraph 2: The phrasing that the rise in e-cigarette use is “contributing to continued to declines in smoking prevalence” is not necessarily the same as an association between more e-cigarette use and less smoking. Would consider providing clear data that the rise in e-cigarette use is what is actually driving smoking rates down. Introduction, Page 5, Paragraph 2: It is mentioned that NRT is used as a comparator because it is a non-combustible nicotine product. Curious why NRT was chosen instead of smokeless tobacco products? Methods, Page 8, Paragraph 1: In the mention of weighting, there is reference to tenure and working status. Does not appear that these were previously defined. Results, Page 8, Paragraph 3: Did any study participants use both NRT and e-cigarettes long-term? Are the groups mutually exclusive or is there overlap? Results, Page 9, Paragraph 3: The word “participants” is missing after “a total of 8649..” Discussion, Page 11, Paragraph 2: Given that many of these analyses were cross-sectional, and that the study looked at current long-term use of e-cigarettes, it does not seem like any conclusions could be drawn about e-cigarettes serving as a gateway to cigarette smoking. Presumably if that was the case, it would be happening after this period of data collection and would not be captured in this cross-sectional analysis. Discussion, Page 11, Paragraph 2: It appears that long-term e-cigarette use is more prevalent in older individuals than younger individuals in this study. Is the prevalence of e-cigarette use in England different by age group? May be a good opportunity to compare/contrast these prevalence rates to other countries. Discussion, Page 11, Paragraph 3: With respect to the fact that people tend to discontinue NRT more quickly after quitting smoking, this is likely reflective of the fact that consumers may view e-cigarettes as more of a substitute for the behavior of cigarette smoking than they would NRT. Potentially could also be
--	--

	considering NRT as more of a “medication” than recreational product. Discussion, Page 12, Paragraph 1: Is there any data on the difference in cost between NRT and e-cigarettes in England? Table 1: Is it possible to include the total study population’s sociodemographic factors here for comparison?
--	--

REVIEWER	Sue Cooper University of Nottingham UK
REVIEW RETURNED	04-Mar-2019

GENERAL COMMENTS	Long-term e-cigarette and nicotine replacement therapy use in England: a population survey I enjoyed reading this interesting study examining the prevalence of long-term e-cigarette use in comparison to long-term NRT use. The authors also examine the characteristics associated with this use. The data came from the well-respected and well-established Smokers Toolkit study, with a large representative sample of English adults. In addition to the cross-sectional data, the study followed up individuals who reported being past-year smokers at baseline after 6 and 12 months. I have a few comments and queries.  1. I wondered why only those who reported past-year smoking at baseline were invited to complete follow-up surveys. I can understand not including never smokers (as they are very unlikely to use e-cigs), but as e-cigarettes have been around for a good while now, there may be people who quit smoking longer than one year ago but still use them, and so the information from these users will have been missed. As demonstrated in Table 1, it appears that being a long-term ex-smoker isn’t that different from being a more recent ex-smoker in terms of long-term e-cigarette use. Perhaps there were reasons for not including these (such as capacity for conducting follow-up), but these don’t appear to be explained. 2. It would be helpful to have some additional information about the prospective part of the study. For example, I don’t think the manuscript mentions how many participants agreed to be followed up, how many calls were made in the attempt to follow them up, and percentage of follow-ups at each wave and overall. In addition the abstract mentions 733 as being the number who were followed up, but this is perhaps slightly misleading as this is the number who responded rather than the number who were followed up. 3. Findings for long-term e-cigarette use in the cross-sectional and prospective parts of the study were very different, and when examined in Table 3, the two samples have quite different baseline characteristics. There are limitations for the prospective data due to very low response rates to the follow-up surveys (which the authors discuss). In Table 3, long-term e-cigarette use is 3.8% in the baseline sample, and 5.5% in the follow-up sample, with a weighted prevalence of 13.4% given in the text for prospectively reported use. Equivalent figures for NRT are 1.3%, 2.6% and 1.9%. This seems a large difference. How
---

	reliable is weighting? I acknowledge that limitations are discussed in the manuscript, but the prevalence findings for the prospective data are still presented as a fairly major part of the study, including in the abstract, with conclusions made that prevalence may be a lot higher than may appear from the cross-sectional findings. Although attrition is mentioned as a limitation, depending on how reliable weighting is, is it possible that by including in the abstract these conclusions may still be overstating the findings? Is this worth discussing further? Would multiple imputation be more reliable than weighting? 4. The finding that e-cigarette users are more likely to continue using these long-term than NRT users is interesting. This aligns with recent findings from Peter Hajek's e-cigarette trial published in NEJM recently where a larger proportion of participants who had quit in the e-cig group were still their assigned product after one year than those in the NRT group. This paper will have been published since this manuscript was written, but it would be a useful addition to make some comments on this study for comparison. Minor comments. 5. Methods p6 lines 33-40. It might be helpful to include bit more information on the questions used. For example, were participants asked if they used the products to help them quit (it only mentions to help them cut down)? Plus, in the question about how long they had been using the product – was there a definition of what was meant by use? For example, did this have to be regular use or could it be occasional (and if so what counted as regular/occasional)?
--	---

REVIEWER	Laurie Zawertailo Centre for Addiction and Mental Health Toronto, Canada
REVIEW RETURNED	14-Mar-2019

GENERAL COMMENTS	MS: 2019-029252 Long-term e-cigarette and NRT use in England: A population survey This paper describes an analysis of data from the Smoking Toolkit, a cross-sectional survey in England conducted between 2016 and 2018. The authors also asked respondents if they could contact them again in 6 and 12-months – these 700+ people comprised the sample for a prospective survey of long-term e-cig and NRT usage. The main finding was that long-term e-cig use was relatively infrequent occurring in only 1.5% of adults and 4% of smokers. Long-term NRT use was even more infrequent which is not surprising. Among adults who were never-smokers, long-term e-cig use is practically non-existent. For the prospective data it appears that long-term e-cig use was more prevalent among this group but the strong selection bias limits the validity of these findings. Overall this is a nicely written paper with some interesting findings. My main concern is the over-statement of the potential significance of the findings from the prospective study. Specific Comments: There is no description of the consenting process for the overall survey nor for the prospective survey. This needs to be included.
--

	As well, there is no mention of who provided ethics oversight for the study. Please include. The AUDIT score is described as a measure of alcohol consumption. This is inaccurate. It is a measure of disordered drinking, not consumption per se. Also, the AUDIT is usually structured as 2 parts – the first 3 questions which comprise the AUDIT-C, are used to screen for potential problematic use. If the score is above the cut-off, then the remaining 3 questions are triggered and asked and a total AUDIT score is calculated. Is this what was done here or were all questions asked regardless of the answers to the first 3? The ‘social grade’ variable seems a bit crude to me. The 2 groupings don’t make intuitive sense. Perhaps a better explanation of how this was decided is warranted. For the regression models how was it decided which variables to include in the model? As I stated before, the data from the prospective survey suffers from extreme self-selection bias and this needs to be stated up front. The authors may want to consider not presenting these data all for this reason.
--	--

VERSION 1 – AUTHOR RESPONSE

Reviewer: 1

The authors are to be commended for a well-written study that anticipates important questions regarding long-term use of e-cigarette-type products (ecigs). They compare use of these products to long-term use of NRT, which provides information about enduring use of nicotine that is not delivered via tobacco. The study is descriptive in nature and the results tell us that long-term ecig use is not (yet) highly prevalent and that, importantly, their use is confined to current and ex-smokers. This is reassuring, and indicates that long-term use may confer some advantages in terms of helping smokers to cut down/quit combustible tobacco use.

Minor point: The authors should describe in a bit more detail why the STS yields a representative sample that permits population-level statistical inferences.

Response: We have added to our description of the study design:

“Data were used from the Smoking Toolkit Study (STS), an ongoing monthly repeat cross-sectional survey of adults in England (22). Each month, a form of random location sampling is used to select a new sample of approximately 1,700 adults aged 16 years and older. Grouped output areas (containing ~300 households) are stratified by ACORN (socio-demographic) characteristics (<http://www.caci.co.uk/acorn/acornmap.asp>) and region before being randomly selected for inclusion in an interviewers list. Interviewers then choose which houses within these areas are most likely to fulfil their quotas and conduct face-to-face computer-assisted interviews with one member per household. Comparisons of sociodemographic data and smoking prevalence and consumption estimates with national data indicate that STS data are broadly representative of the English population, having a similar composition to other large national surveys, such as the Health Survey for England (22).”

In addition, in a paper currently under review, we have compared estimates of cigarette consumption in the Smoking Toolkit Study with cigarette sales data for England, finding very close alignment.

Reviewer: 2

This manuscript evaluates the prevalence of long-term e-cigarette and nicotine replacement therapy use among adults in England, and factors associated with long-term use of each product. The authors found a higher prevalence of long-term e-cigarette use than NRT use, both among all adults and among current smokers. They defined long-term product use in two different ways – retrospectively among all participants and prospectively among smokers – and found higher prevalence rates with the prospective definition, although the latter estimation had the limitation of potential bias from loss to follow-up and a difference in definition. Strengths of this study include the large sample size and representative sample. Limitations involve contextual factors as noted below. I do think that the paper would be strengthened by a comparison between long-term users of e-cigarettes/NRT and short-term users of e-cigarettes/NRT in this specific population for context; based on how e-cigarette use was assessed in the survey, it seems that this should be possible.

Response: We have added comparative statistics on current (as opposed to long-term) use in the discussion, as these have already been published elsewhere – please see response to your later comment.

Abstract, Setting: Would include the year(s) of data collection here.

Response: We have added this to the design section:

“Design: Cross-sectional and prospective survey, the Smoking Toolkit Study, with baseline data collected between September 2014 and September 2016 and follow-ups at 6 and 12 months.”

Abstract, Results: The reference to “Long-term e-cigarette and NRT use” sounds like it is referring to people who use both e-cigarettes and NRT. Would recommend rephrasing.

Response: We have edited to “Long-term e-cigarette use and long-term NRT use”.

Introduction, General: It would be helpful to have some context on the regulatory/marketing environment in the England for non-UK readers. Are both e-cigarettes and NRT available over the counter? Are they available by prescription? I.e. how easy is it to access each product

Response: We have added the following to the introduction:

“In England, e-cigarettes are not currently available on prescription but are subject to the EU Tobacco Products Directive (including advertising restrictions) and Trading Standards and can be bought online and from vape shops, pharmacies and other retail outlets, while NRT can be bought over the counter or obtained on prescription from a licensed health professional.”

Introduction, Page 4, Paragraph 2: Would add the 2019 RCT in NEJM by Hajek et al to the two RCTs mentioned here.

Response: Thank you, this study was not published when we wrote this paper. We now refer to it in the introduction.

Introduction, Page 4, Paragraph 2: The phrasing that the rise in e-cigarette use is “contributing to continued declines in smoking prevalence” is not necessarily the same as an association between more e-cigarette use and less smoking. Would consider providing clear data that the rise in e-cigarette use is what is actually driving smoking rates down.

Response: We have tempered the language here. It now reads:

“On a population level, the rise in use of e-cigarettes in England and the US has been associated with increases in the overall success rate of quit attempts in the population (10,11), likely contributing to continued declines in smoking prevalence (12).”

Introduction, Page 5, Paragraph 2: It is mentioned that NRT is used as a comparator because it is a non-combustible nicotine product. Curious why NRT was chosen instead of smokeless tobacco products?

Response: We considered NRT to be a more appropriate comparison than smokeless tobacco because, like e-cigarettes, (i) it is non-tobacco nicotine product, and (ii) it is widely used in the population. There would not have been sufficient numbers to analyse long-term use of smokeless tobacco products due to low prevalence of use in England, given the sale of many forms (e.g. snus) is banned by the EU in England.

Methods, Page 8, Paragraph 1: In the mention of weighting, there is reference to tenure and working status. Does not appear that these were previously defined.

Response: We now define these variables:

“Rim (marginal) weighting was used to match the English population on the dimensions of age, social grade, region, housing tenure (bought on a mortgage, owned outright, rented from local authority, rented from private landlord), ethnicity and working status (working, not working) within sex.”

Results, Page 8, Paragraph 3: Did any study participants use both NRT and e-cigarettes long-term? Are the groups mutually exclusive or is there overlap?

Response: We now clarify in the method:

“Participants who reported long-term use of both e-cigarettes and NRT (n=66) were excluded.”

Results, Page 9, Paragraph 3: The word “participants” is missing after “a total of 8649..”

Response: We have corrected this.

Discussion, Page 11, Paragraph 2: Given that many of these analyses were cross-sectional, and that the study looked at current long-term use of e-cigarettes, it does not seem like any conclusions could be drawn about e-cigarettes serving as a gateway to cigarette smoking. Presumably if that was the case, it would be happening after this period of data collection and would not be captured in this cross-sectional analysis.

Response: We do not comment on whether or not there is a true causal gateway, and that the reviewer is correct that the current study design can't examine this. Our rationale for raising this here is that the very low prevalence of long-term e-cigarette use among never smokers means - even if e-cigarette use did cause never smokers to become smokers - then the potential the number of never smokers taking up smoking as a result of e-cigarette use is likely to be very small. This is a comment only on the situation as we have observed in England. We have clarified this now:

“Long-term use of both products was almost exclusively observed among current and former smokers. Concerns have been raised that e-cigarettes may serve as a gateway to cigarette smoking among never-smokers, particularly among youth (30,31), but in our sample, the prevalence of long-term e-cigarette use among never-smokers was just 0.1%, comparable to long-term NRT use. As such, the potential number of people susceptible to be any gateway effects in England between 2014 and 2016 appears to have been very small.”

Discussion, Page 11, Paragraph 2: It appears that long-term e-cigarette use is more prevalent in older individuals than younger individuals in this study. Is the prevalence of e-cigarette use in England different by age group? May be a good opportunity to compare/contrast these prevalence rates to other countries.

Response: We have added some comparative statistics on current use of e-cigarettes by age group in England that have been published elsewhere. These figures are also based on Smoking Toolkit Study data:

“We also observed higher prevalence of long-term e-cigarette use among middle-aged and older adults than in the youngest group (16-34 years), in contrast with evidence that current use of e-cigarettes among current and former smokers in England is least prevalent in the oldest age group (12.2% in those aged ≥ 65 years, compared with 18.7%, 21.4%, 20.8%, 20.6%, and 18.6% in those aged 16-24, 25-34, 35-44, 45-54, and 55-64 years) (7).”

Discussion, Page 11, Paragraph 3: With respect to the fact that people tend to discontinue NRT more quickly after quitting smoking, this is likely reflective of the fact that consumers may view e-cigarettes as more of a substitute for the behavior of cigarette smoking than they would NRT. Potentially could also be considering NRT as more of a “medication” than recreational product.

Response: We have added these potential explanations:

“This suggests that people tend to discontinue use of NRT more quickly after quitting smoking than with e-cigarettes, possibly because e-cigarettes are a closer substitute for the behaviour of cigarette smoking than NRT, or because NRT is viewed more as a medication than a recreational product (9).”

Discussion, Page 12, Paragraph 1: Is there any data on the difference in cost between NRT and e-cigarettes in England?

Response: We currently have a revised manuscript under review at Addiction describing differences in expenditure on NRT and e-cigarettes by users of these different product categories, which we now cite here:

“According to this theory, one would expect to see e-cigarette use spread first within more affluent social networks, but patterns of imitation later lead to diffusion of the practice and normative change across the socio-economic range. The fact that we observed no significant association between education or social grade and long-term NRT use, which has a similar cost to users (40) but has been available for much longer, is consistent with this.”

Table 1: Is it possible to include the total study population’s sociodemographic factors here for comparison?

Response: We have added this information in an additional column.

Reviewer: 3

I enjoyed reading this interesting study examining the prevalence of long-term e-cigarette use in comparison to long-term NRT use. The authors also examine the characteristics associated with this use. The data came from the well-respected and well-established Smokers Toolkit study, with a large representative sample of English adults. In addition to the cross-sectional data, the study followed up individuals who reported being past-year smokers at baseline after 6 and 12 months. I have a few comments and queries.

1. I wondered why only those who reported past-year smoking at baseline were invited to complete follow-up surveys. I can understand not including never smokers (as they are very unlikely to use e-cigs), but as e-cigarettes have been around for a good while now, there may be people who quit smoking longer than one year ago but still use them, and so the information from these users will have been missed. As demonstrated in Table 1, it appears that being a long-term ex-smoker isn’t that different from being a more recent ex-smoker in terms of long-term e-cigarette use. Perhaps there were reasons for not including these (such as capacity for conducting follow-up), but these don’t appear to be explained.

Response: The follow-up schedule was determined prior to this specific analysis being designed. As you suggest, this is primarily driven by cost implications (following all ever smokers would drive up costs substantially) but also because relapse rates after one year of decrease substantially. Given the original focus of STS to look at KPI for smoking and smoking cessation, we’re most interested in what happens immediately after a quit attempt to improve long-term abstinence rates (so are less interested in those who have successfully transitioned into long-term abstinence).

We have now expanded on our discussion of this as a limitation in the discussion:

“Only respondents who reported past-year smoking at baseline were invited to participate in the follow-up surveys, so we were unable to obtain prospective estimates of the prevalence of long-term

e-cigarette or NRT use in the entire adult population. While, evidence from the cross-sectional results of this study and from previous research (3) suggest that the vast majority of long-term users were current or recent ex-smokers, with low prevalence among never smokers, it would have been useful to have data from long-term ex-smokers.”

2. It would be helpful to have some additional information about the prospective part of the study. For example, I don't think the manuscript mentions how many participants agreed to be followed up, how many calls were made in the attempt to follow them up, and percentage of follow-ups at each wave and overall. In addition the abstract mentions 733 as being the number who were followed up, but this is perhaps slightly misleading as this is the number who responded rather than the number who were followed up.

Response: We have added the following to the method: “Up to 7 attempts were made to follow up each consenting participant.” We have edited the abstract to read: “...in a subsample of smokers who responded to follow-up (n=733).”

3. Findings for long-term e-cigarette use in the cross-sectional and prospective parts of the study were very different, and when examined in Table 3, the two samples have quite different baseline characteristics. There are limitations for the prospective data due to very low response rates to the follow-up surveys (which the authors discuss). In Table 3, long-term e-cigarette use is 3.8% in the baseline sample, and 5.5% in the follow-up sample, with a weighted prevalence of 13.4% given in the text for prospectively reported use. Equivalent figures for NRT are 1.3%, 2.6% and 1.9%. This seems a large difference. How reliable is weighting? I acknowledge that limitations are discussed in the manuscript, but the prevalence findings for the prospective data are still presented as a fairly major part of the study, including in the abstract, with conclusions made that prevalence may be a lot higher than may appear from the cross-sectional findings. Although attrition is mentioned as a limitation, depending on how reliable weighting is, is it possible that by including in the abstract these conclusions may still be overstating the findings? Is this worth discussing further? Would multiple imputation be more reliable than weighting?

Response: We appreciate this suggestion, and have added a sensitivity analysis using multiple imputation for prospective prevalence estimates:

Method: “Following peer review, we added an unplanned sensitivity analysis of the prospective data in which missing data on e-cigarette and NRT use at 6 months and 12 months were imputed for all baseline past-year smokers with missing data. We used a multiple imputation model with all baseline sociodemographic and smoking-related characteristics, baseline use of e-cigarettes, and baseline use of NRT as predictors. Five imputed datasets were created, each analysed separately, and the results combined to produce pooled estimates of prevalence of long-term use of e-cigarettes and long-term use of NRT.”

Results: “When missing data on use of e-cigarettes and NRT at 6 and 12 months were multiply imputed for participants who were past-year smokers at baseline and did not participate in the follow-up surveys (n=1,673, 69.5% of all baseline past-year smokers), the unweighted prevalence of long-term e-cigarette use assessed prospectively was 9.8% (95%CI 9.2-10.4%) and of long-term NRT use was 1.7% (95%CI 1.4-2.0%), and the weighted prevalence was 10.3% (95%CI 9.7-11.0%) and 1.6% (95%CI 1.3-1.9%), respectively.”

Discussion: “Only a minority of past-year smokers retrospectively reported long-term use of either e-cigarettes (3.9%) or NRT (1.3%) but this figure may be an underestimate: prevalence of current use at three time-points over a 12-month period was substantially higher for both e-cigarettes (13.4%) and NRT (1.9%), although these estimates were likely subject to attrition bias. When missing data were imputed, prospectively assessed prevalence estimates were slightly lower, at 10.3% for long-term e-cigarette use and 1.6% for long-term NRT use.”

4. The finding that e-cigarette users are more likely to continue using these long-term than NRT users is interesting. This aligns with recent findings from Peter Hajek's e-cigarette trial published

in NEJM recently where a larger proportion of participants who had quit in the e-cig group were still their assigned product after one year than those in the NRT group. This paper will have been published since this manuscript was written, but it would be a useful addition to make some comments on this study for comparison.

Response: Thank you for this suggestion. We have added the following:

“A recent trial of e-cigarettes compared with NRT in UK stop smoking services observed similar, with participants randomised to use e-cigarettes in a quit attempt more likely than those randomised to use NRT to still be using their allocated product one year later (80% vs. 9%, respectively) (9).”

Minor comments.

5. Methods p6 lines 33-40. It might be helpful to include bit more information on the questions used. For example, were participants asked if they used the products to help them quit (it only mentions to help them cut down)? Plus, in the question about how long they had been using the product – was there a definition of what was meant by use? For example, did this have to be regular use or could it be occasional (and if so what counted as regular/occasional

Response: We now provide the exact wording of these questions:

“In each of the baseline and follow-up surveys, three questions asked respondents about current use of e-cigarettes (or other vaping devices) and NRT (e.g. nicotine patches, gum, spray, or any other product):

1. Which, if any, of these are you currently using to help you cut down the amount you smoke?
2. Do you regularly use either of these in situations when you are not allowed to smoke?
3. Can I check, do you currently use either of the following at all for any reason?

“In the baseline survey, respondents reporting use of either e-cigarettes or NRT were asked: “How long have you been using this nicotine replacement product or these products for?” Response options were: (i) less than one week, (ii) one to six weeks, (iii) more than six weeks and up to 12 weeks, (iv) more than 12 weeks and up to 26 weeks, (v) more than 26 weeks and up to 52 weeks, and (vi) more than 52 weeks.”

Reviewer: 4

Long-term e-cigarette and NRT use in England: A population survey This paper describes an analysis of data from the Smoking Toolkit, a cross-sectional survey in England conducted between 2016 and 2018. The authors also asked respondents if they could contact them again in 6 and 12-months – these 700+ people comprised the sample for a prospective survey of long-term e-cig and NRT usage. The main finding was that long-term e-cig use was relatively infrequent occurring in only 1.5% of adults and 4% of smokers. Long-term NRT use was even more infrequent which is not surprising. Among adults who were never-smokers, long-term e-cig use is practically non-existent. For the prospective data it appears that long-term e-cig use was more prevalent among this group but the strong selection bias limits the validity of these findings.

Overall this is a nicely written paper with some interesting findings. My main concern is the overstatement of the potential significance of the findings from the prospective study.

Specific Comments:

There is no description of the consenting process for the overall survey nor for the prospective survey. This needs to be included.

Response: We have added this to the description of the design and study population:

“All participants provide fully informed consent prior to participation. In each wave, respondents complete a face-to-face computer-assisted survey with a trained interviewer. Respondents to the baseline survey between September 2014 and September 2016 who reported smoking in the past

year were asked whether they were willing to be re-contacted, and those who agreed were followed up by telephone 6 and 12 months after the baseline interview.”

As well, there is no mention of who provided ethics oversight for the study. Please include.

Response: The following is included in the Declarations section:

“Ethical approval for the Smoking Toolkit Study was granted originally by the UCL Ethics Committee (ID 0498/001) and participants provided full informed consent. The data are not collected by UCL and are anonymised when received by UCL.”

The AUDIT score is described as a measure of alcohol consumption. This is inaccurate. It is a measure of disordered drinking, not consumption per se. Also, the AUDIT is usually structured as 2 parts – the first 3 questions which comprise the AUDIT-C, are used to screen for potential problematic use. If the score is above the cut-off, then the remaining 3 questions are triggered and asked and a total AUDIT score is calculated. Is this what was done here or were all questions asked regardless of the answers to the first 3?

Response: We appreciate this concern and have edited our wording throughout the manuscript to make it clear that the measure was of high-risk drinking, rather than alcohol consumption per se. The items were administered as you describe.

The ‘social grade’ variable seems a bit crude to me. The 2 groupings don’t make intuitive sense. Perhaps a better explanation of how this was decided is warranted.

Response: This measure is a widely used and validated indicator of socioeconomic position in the UK population. We have provided more information to support the validity of the measure in the methods section: “This occupational measure of social grade is a valid index of SES that is widely used in research in UK populations. It has been identified as particularly relevant in the context of tobacco use and quitting (23) and other addictive behaviours (24). These social grades are frequently amalgamated into two groupings; ABC1 and C2DE. Here, researchers frequently interpret ABC1 to represent the middle class and C2DE to represent the working class.”

For the regression models how was it decided which variables to include in the model?

Response: All variables were selected a priori. We now explain this in the method:

“Data were included on a range of sociodemographic and smoking-related characteristics assessed at baseline, selected a priori on the basis of previous studies demonstrating associations with use of e-cigarettes and/or NRT.”

As I stated before, the data from the prospective survey suffers from extreme self-selection bias and this needs to be stated up front. The authors may want to consider not presenting these data all for this reason.

Response: We highlight this as one of the study’s key limitations:

“Substantial attrition bias meant our sample for prospective analyses was older and more socioeconomically advantaged than the group who were lost to follow-up, and more reported recent quitting and long-term use of e-cigarettes or NRT retrospectively at baseline.”

And again in the discussion:

“Another potential issue was substantial attrition bias. Our sample for prospective analyses was older and more socioeconomically advantaged than the group who were lost to follow-up, and more reported long-term use of e-cigarettes or NRT retrospectively at baseline. They were also more likely to have quit recently. This may have contributed to the higher prevalence of long-term use observed in prospective analyses.”

VERSION 2 – REVIEW

REVIEWER	Sara Kalkhoran Massachusetts General Hospital/Harvard Medical School, USA
REVIEW RETURNED	13-Aug-2019

GENERAL COMMENTS	Thank you for the opportunity to review this revised manuscript. The authors have done a nice job addressing the concerns raised during peer review. I only have a few comments: Methods, Page 8, Lines 14-21: I am curious why cigarettes per day was not included as a smoking-related characteristic. Certainly that could influence use of other forms of nicotine. Discussion, Page 12, Lines 22-27: Are there any thoughts as to why anyone who has never smoked uses NRT? It is a very small group (n=7), but given that there are multiple parallels being drawn to never smokers who use e-cigarettes, would be curious to know the authors' thoughts. Furthermore, while it is mentioned that the potential number of people susceptible to gateway effects is small, people who use e-cigarettes for <1 year could still be susceptible to later start smoking cigarettes. Without data on individuals who use e-cigarettes for <1 year, this can't be properly clarified. Table 2: Would be helpful to clarify how many people were included in each of the multivariable models or if there was any missing data for the variables that were included.
---

REVIEWER	Sue Cooper University of Nottingham, United Kingdom
REVIEW RETURNED	28-Jun-2019

GENERAL COMMENTS	I am happy with the amendments made by the authors and have no further comments. (Typing error on last paragraph of the results: multiply rather than multiple)
--

REVIEWER	Laurie Zawertailo Centre for Addiction and Mental Health Toronto, CANADA
REVIEW RETURNED	26-Jun-2019

GENERAL COMMENTS	Thank you for addressing all of my comments
---

VERSION 2 – AUTHOR RESPONSE

Reviewer: 2

Reviewer Name: Sara Kalkhoran

Institution and Country: Massachusetts General Hospital/Harvard Medical School, USA Please state any competing interests or state 'None declared': None declared

Please leave your comments for the authors below Thank you for the opportunity to review this revised manuscript. The authors have done a nice job addressing the concerns raised during peer review. I only have a few comments:

Methods, Page 8, Lines 14-21: I am curious why cigarettes per day was not included as a smoking-related characteristic. Certainly that could influence use of other forms of nicotine.

Response: We included time to first cigarette (TTFC) and believed it likely that CPD would be collinear. TTFC is regarded as the best single indicator of cigarette dependence. Also, CPD is changing over time in countries with strong tobacco control measures, which makes its inclusion in analyses over time complicated.

Baker TB, Piper ME, McCarthy DE, Bolt DM, Smith SS, Kim S, Colby S, Toll BA. Time to first cigarette as an index of ability to quit smoking: implications for nicotine dependence. *Nicotine & Tobacco Research*. 2007;9:S555–S570

Fagerstrom K. Time to first cigarette; the best single indicator of tobacco dependence? *Monaldi Archives for Chest Disease*. 2003 59:91–94.

Martin J. Jarvis, Gary A. Giovino, Richard J. O'Connor, Lynn T. Kozlowski, John T. Bernert, Variation in Nicotine Intake Among U.S. Cigarette Smokers During the Past 25 Years: Evidence From NHANES Surveys, *Nicotine & Tobacco Research*, Volume 16, Issue 12, December 2014, Pages 1620–1628,

Kuipers MA, Partos T, McNeill A, et al Smokers' strategies across social grades to minimise the cost of smoking in a period with annual tax increases: evidence from a national survey in England *BMJ Open* 2019;9:e026320. doi: 10.1136/bmjopen-2018-026320

Discussion, Page 12, Lines 22-27: Are there any thoughts as to why anyone who has never smoked uses NRT? It is a very small group (n=7), but given that there are multiple parallels being drawn to never smokers who use e-cigarettes, would be curious to know the authors' thoughts.

Response: It is an interesting question. For context, NRT has been widely available over-the-counter in England for a long period of time, including fast-acting forms such as gum and nasal spray, which are effective nicotine delivery mechanisms. Given the size of the group in the English population, a substantial absolute number of long-term ex-smokers use NRT.

The explanation is probably as complex and multi-faceted as the question why do people smoke or use e-cigarettes? Experimentation likely relates to beliefs about the benefits and enjoyment of nicotine, exposure to peers, health concerns and so on; persistence probably relates to heritability of nicotine dependence, harm perceptions and a variety of other factors related to the opportunity, motivation and capability for using NRT. Ultimately, those who have used for an extended period are likely to continue using because they have failed to stop successfully, which is likely related primarily to their nicotine dependence.

Elsewhere we have reported current use (including but not limited to long-term use) by smoking status and also find a small (~0.5%) proportion of never smokers use NRT. In both cases, the figures are a similar order of magnitude but marginally less than for e-cigarettes in England.

West R. (2009) The Multiple Facets of Cigarette Addiction and What They Mean for Encouraging and Helping Smokers to Stop, COPD: Journal of Chronic Obstructive Pulmonary Disease, 6:4, 277-283, DOI: 10.1080/15412550903049181

Furthermore, while it is mentioned that the potential number of people susceptible to gateway effects is small, people who use e-cigarettes for <1 year could still be susceptible to later start smoking cigarettes. Without data on individuals who use e-cigarettes for <1 year, this can't be properly clarified.

Response: We have edited our wording to clarify that we are only talking about long-term e-cigarette use:

“As such, the potential number of people susceptible to any gateway effects as a result of long-term e-cigarette use in England between 2014 and 2016 appears to have been very small.”

Table 2: Would be helpful to clarify how many people were included in each of the multivariable models or if there was any missing data for the variables that were included.

Response: We now clarify in the description of the statistical analyses in the method:

“All analyses were done in SPSS v.25 on complete cases.”

Numbers of participants included in the analyses reported in Tables 1 and 2 are reported in the table titles.

VERSION 3 – REVIEW

REVIEWER	Sara Kalkhoran Massachusetts General Hospital/Harvard Medical School
REVIEW RETURNED	18-Sep-2019
GENERAL COMMENTS	Thank you for your response to my comments.